# Tokenization of the Common:
# An Economic Model of Multidimensional Incentives

**Benjamin Kraner**
Blockchain & Distributed Ledger
Technology Research Group
Department of Informatics
University of Zurich
Zürich, Switzerland
benjamin.kraner@uzh.ch

**Nicolò Vallarano**
UZH Blockchain Center
University of Zurich
Zürich, Switzerland

**Claudio J. Tessone**
UZH Blockchain Center
University of Zurich
Zürich, Switzerland

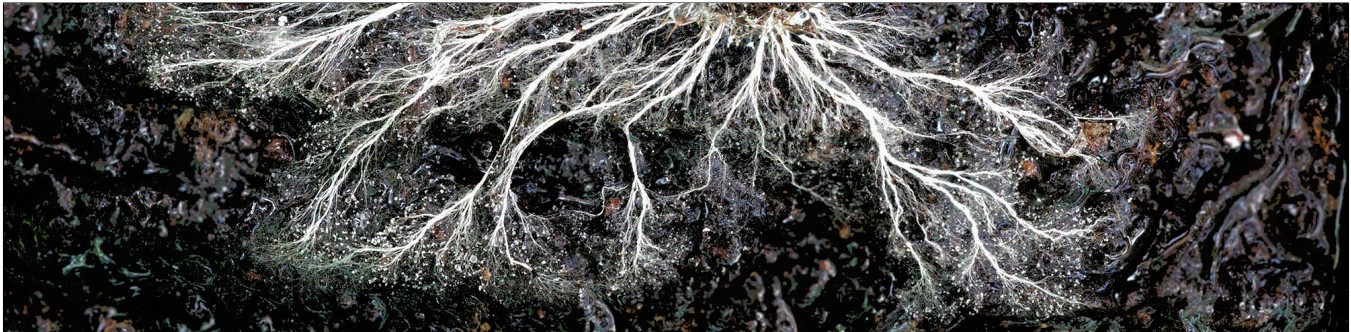

**Figure 1.** Mirror of nature: Our research, inspired by fungi networks that balance multiple inputs like oxygen and nitrogen, mirrors this complexity in a tokenized system, where each goal and good is represented by a unique token.[*]

## Abstract

The concept of the tragedy of the commons, originally rooted in economics, describes the depletion of shared resources due to self-interested actions by individuals. This work proposes a novel solution to address this economic challenge by leveraging tokens to capture its multidimensional nature. By utilising blockchain and DLTs, this decentralised approach aims to achieve a social optimum while promoting self-regulation. The paper presents a mathematical treatment of the tragedy of the commons, incorporating multi-dimensional tokens and exploring the divergence from the classic optimal solution, highlighting the potential of tokenisation in shaping a sustainable and efficient economy.

*CCS Concepts:* • **Networks** → *Peer-to-peer protocols*; *Peer-to-peer protocols*; • **Information systems** → Distributed storage; Distributed storage; • **Security and privacy** → Economics of security and privacy; Economics of security and privacy.

*Keywords:* Tragedy of the Commons, Shared Resource, Tokens, Decentralised solution, Social Optimum, Blockchain technology, Decentralised Ledger Technology (DLT), Smart Contracts

*DICG'23, December 11–15, 2023, Bologna, Italy*
© 2023 Association for Computing Machinery.
ACM ISBN 978-x-xxxx-xxxx-x/YY/MM...$15.00
https://doi.org/3631310.3633486

**ACM Reference Format:**
Benjamin Kraner, Nicolò Vallarano, and Claudio J. Tessone. 2023. Tokenization of the Common: An Economic Model of Multidimensional Incentives. In *Proceedings of 4th International Workshop on Distributed Infrastructure for Common Good (DICG'23)*. ACM, New York, NY, USA, 6 pages. https://doi.org/3631310.3633486

## 1 Introduction

The tragedy of the commons [8] refers to a situation where multiple individuals have access to a shared resource, such as a fishery, a water source or a pasture, and each individual acts to maximise their own consumption of that resource. Over time, this leads to overuse and depletion of the resource, even though it is held in common and intended to be shared among all users. A common solution proposed to address the tragedy of the commons is the imposition of property rights and market-based mechanisms, such as the creation of tradable pollution permits or the introduction of a quota system for the use of a shared resource. A classic approach is the application of game theory to analyse the incentives and behaviour of individuals in the context of a shared resource[3, 5]. This has led to the development of the commons dilemma, a non-cooperative game that models the interactions between individuals and their impact on the shared resource. Studies have also explored the role of institutions and governance mechanisms in mitigating the tragedy of the commons, including the use of community-based management, cooperation between stakeholders, and the enforcement of regulations and sanctions [2, 12].

Our approach follows a different path. In recent years, we witnessed the proliferation and development of blockchain technology

[*]Image-Source: Creator: Danny Newman (2015), Licensed under CC BY-NC-ND 4.0 DEED. Downloaded from https://www.thisiscolossal.com/wp-content/uploads/2018/01/fungi-1.jpg on October 4th, 2023

[13] — or in general, decentralised ledger technology (DLT). In the specific, the development of smart contracts[11] — self-executing digital contracts with the terms of the agreement between parties directly written into code — transformed the way people interact with money by offering an alternative to traditional financial systems. Smart contracts can be used to create tokens which can be bought, sold, or traded. This can lead to the creation of a fully tokenized economy [9] where all types of assets can be traded on blockchain platforms, sustained by the open-source nature of smart contracts that implies users can design, implement and share their incentive schemes freely.

The use of tokens enables individuals to generate and exchange value beyond the economic dimension. Consider, as a reference, the reputation systems employed on social media sites like StackOverflow and Reddit, known as *reputation* and *karma*, respectively. Although these systems might yield economic returns [7], they capture a value that is not purely economic, such as the interactions between Reddit users collecting *mana*. Tokenization makes such value tradable and recognisable on broader platforms, an element crucial for the modern digital society.

Tokenization can be instrumental in scenarios where it may be difficult to assign an economic value to items that nonetheless have use and value within a community, due to its capacity to encapsulate non-economic value[4]. Another example of this is the climate crisis, a typical case of application of the tragedy of the commons. In this context, we already see government agencies pushing research on DLT applications to scale up climate action, such as *COP28 UAU*[1].

## 1.1 Motivations and Goals

At the crossroads of classical economics and decentralised ledger technologies, with the present work we present a modelization of the use of tokens to develop self-regulatory communities, where ad-hoc incentives would drive individual participants to follow positive strategies for the collective system. In practice, we present an alternative solution to the tragedy of the commons relying on multi-dimensional incentives rather than a social planner.

By relying on the digital infrastructure provided by decentralised ledger technologies and blockchains [13, 14] our approach has the benefit of being fully decentralised, as opposed to traditional solutions of the tragedy of the commons problem that usually requires the presence of a central party, i.e. a state institution or a middle-man firm, allowing for true self-regulation. Additionally, this could be achieved in a fully digital manner, with the clear advantages of being waste-free, infinitely reproducible and with cheap bootstrapping costs for real-world applications.

In the following sections, we will present a simple mathematical treatment and of the tragedy of the commons with the introduction of multi-dimensional tokens, with particular attention on the divergence from the classic optimal solution. Then, we will present a stylised example of a possible application. At last, we will discuss the results and further directions.

## 2 A simple economical model of multidimensional incentives

To lay the groundwork for our discussion, we begin by exploring a *simplified rendition of the tragedy of the Commons* in a *microeconomic framework*. Subsequently, we will *adapt the model* to accommodate secondary tokens, which include elements such as *cryptocurrencies*, *accounting ledgers*, and *reputation systems*.

We aim to present a stylised interpretation of the tragedy of the commons, underscoring the over-exploitation of a resource. This overuse is depicted by excessive individual effort resulting in amplified overall costs. Broadly, this scenario could be applicable to any issue that can be comprehended as a commons problem, where overinvestment diminishes the socially optimal output level. Such issues may be immediate or more long-term, examples of which include overfishing, deforestation, climate change, and so forth. To simplify our model and enhance its tractability, we will disregard the time dimension and concentrate on a single-period model. Further we will assume for the sake of simplicity a completely symmetric problem with homogenous agents. Our proposed model draws inspiration from seminal works in the field. Specifically, we built upon the concepts presented by [10], [6] and [5].

### 2.1 A Simple Common Resource Model

Consider a common resource shared among $N$ individuals. Each individual $i$ consumes an amount $x_i$ of the resource, resulting in total consumption $X$ given by:

$$X = \sum_{i=1}^{N} x_i$$

Assume that the benefit for each individual $i$, denoted as $B_i$, from their consumption $x_i$ is a function $B(x_i)$, and the cost to each individual, denoted as $C$, from the total consumption $X$ is a function $C(X)$. Notably, each individual's cost depends on the total consumption, not their individual consumption level. Therefore, the net utility for each individual $i$, denoted as $v_i$, is given by:

$$v_i = B(x_i) - C(X) = B(x_i) - C(\sum_{j=1}^{N} x_j) \qquad (1)$$

Each individual chooses their consumption $x_i$ to maximize their net utility $v_i$. However, because the cost $C(X)$ depends on the total consumption $X$, each individual's optimal consumption is influenced by the consumption of all other individuals.

#### 2.1.1 Individual Problem.
An individual optimises their utility through the following problem:

$$\max_{x_i} v_i(x_i) = B(x_i) - C(X) \qquad (2)$$

Assuming that $B(x_i)$ is increasing and concave, and $C(X)$ is increasing and convex, a unique Nash equilibrium can be identified. In this equilibrium, each individual selects their consumption $x_i$ such that the marginal benefit equals the marginal cost.

$$B'(x_i) = C'(X)$$
$$= \frac{\partial C}{\partial X} \frac{\partial X}{\partial x_i} = \frac{\partial C}{\partial X} \qquad (3)$$

Given that each agent faces the same optimization problem, we can exploit symmetry in the equilibrium to rewrite eq. (3) as:

$$B'(x_i^*) = C'(N \cdot x_i^*) \qquad (4)$$

---
[1]https://www.cop28.com/en/cop28-uae-techsprint

where $x_i^*$ denotes the optimal solution to the individual's utility maximization problem.

The primary challenge here lies in the externalities introduced by the shared cost among the participants of the economy. While consumption is private, the cost is public. Consequently, agents increase their consumption until their marginal utility gain equals the marginal cost they face, not recognising that they are reducing the system's overall utility and that of other participants. A real-world example of this scenario is overfishing, where excessive fishing by individuals depletes the shared resource, making it more difficult for all participants, including themselves, to fish.

**2.1.2 Social Planner Problem.** However, the socially optimal consumption level, which maximizes the total utility of *all* individuals, is achieved when the sum of the total marginal benefits equals the total marginal cost. A social planner solves:

$$\max_{x_i}(W^{SP}(x_i)) = \max_{x_i} N \cdot v_i(x_i) \tag{5}$$

The social planner can actually enforce the symmetry onto the problem, by imposing that all individuals have an equal share of consumption. Practically, this means $x_i = x^{SP} \ \forall i$ So, by the first-order condition, the optimum is reached at:

$$\sum_{i=1}^{N} B'(x^{SP}) = \sum_{i=1}^{N} \frac{\partial C}{\partial X} \frac{\partial (Nx)}{\partial x}$$
$$= \sum_{i=1}^{N} N \cdot C'(Nx^{SP}) \tag{6}$$

The social planner, therefore, understands the impact of a single individual's consumption increase on society as a whole and corrects for the externalities created by private consumption.

**2.1.3 Overconsumption.** A comparison between the consumption level of the individual and that imposed by the social planner reveals that the common resource is being overused. We can demonstrate this as follows:

$$x_i^* > x_i^{SP}. \tag{7}$$

Since both the social planner and the individual agent adhere to their respective first-order conditions, we can write:

$$B'(x_i^{SP}) - N \cdot C'(x_i^{SP}) = B'(x_i^*) - C'(x_i^*) = 0 \tag{8}$$

Given that $C$ is a monotonically increasing function, where $C' \geq 0$, it follows that:

$$B'(x_i^{SP}) - C'(x_i^{SP}) \geq B'(x_i^{SP}) - N \cdot C'(x_i^{SP}) \tag{9}$$

And, because of eq. (8), we can assert:

$$B'(x_i^{SP}) - C'(x_i^{SP}) \geq B'(x_i^*) - C'(x_i^*) = 0 \tag{10}$$

Now, considering that $B - C$ is a concave function, the overconsumption as described in eq. (7) follows from eq. (10).

In the tragedy of the commons, the Nash equilibrium is not socially optimal, as the sum of the marginal benefits exceeds the marginal cost. Each individual is incentivized to consume more than the socially optimal level, leading to over-exploitation of the common resource. Conversely, the social planner's solution optimizes overall utility by avoiding overconsumption, thus preventing resource depletion and mitigating the negative effects of individual self-interest.

**2.1.4 Assumptions.** We will make a key assumption about how individuals evaluate utility. We propose that utility is not derived solely from money, or more broadly, from goods and services (i.e., everything that money can buy). We argue that there is a component of utility originating from various states of the world which cannot be captured in a strictly monetary sense. In order to substantiate our assumption, we provide an illustrative example. Consider the case of climate change, a prevalent issue with substantial public awareness [1]. Consequently, political will has evolved to address the problem; for instance, an emissions market has been established by the European Union to manage the quantity of $CO_2$ released into the atmosphere. We argue that such developments are manifestations of people's will, or in economic terms, a component of utility that cannot be directly translated into pecuniary terms. We assert that these elements hold value, as demonstrated by revealed preferences, but they cannot be directly priced in simple monetary terms. Current monetary and market systems are unable to accurately capture and convert the necessary information into prices and goods. What we mean by this is that there are no market goods that can effectively combat climate change or halt over-fishing. These issues cannot be commodified within the realm of real money, possibly due to the complexity of the information involved.

Secondly, we assume that there exists a method enabling the measurement and quantification of the system's information and the impact of actions upon it. We propose that each action in the system carries with it a piece of information reflecting its impact towards the desired goal (for instance, the social planner's optimum). This information can be used to assess whether a particular action, which we refer to as the consumption level $x_i$ in our model, aligns with the revealed preferences (i.e., the goal state of the global system).

**2.1.5 The Tokenization of the Common.** In light of our two assumptions, we contend that it is possible to construct a representative token that mirrors the impact of an action on the commons. This can be integrated into our model by defining a function $w(x_i)$. Essentially, we convert the action, specifically the consumption level undertaken by an agent, into a quantifiable metric represented in these tokens.

To clarify our motivation for such a token-based solution, let us consider a motivating example. Suppose we aim to reduce carbon emissions, and an agent faces two options: *taking the train*, associated with low carbon emissions, or *using a personal car*, associated with high carbon emissions. A system could be put in place that recognizes both actions and rewards a varying number of tokens to the agent—more tokens for choosing the train and fewer for selecting the car.[2]

Building upon our utility model, we define an individual's *total utility* as $u_i$. This utility $u_i$ stems from both the agent's economic endeavours (i.e., consumption of the common), represented by $v_i$, and, according to our first assumption, the perceived value of tokens earned, symbolised by $w_i$. These tokens signify the agent's contribution to conserving the commons. Hence, we express $u_i$ as:

$$u_i = v_i(x_i) + w_i(x_i) \tag{11}$$

---

[2]There are several such incentives. For instance, Codos is an initiative that uses blockchain to incentivize environmentally friendly commuting by rewarding sustainable transportation choices with tokens, aiming to reduce $CO_2$ emissions from combustion engine cars.

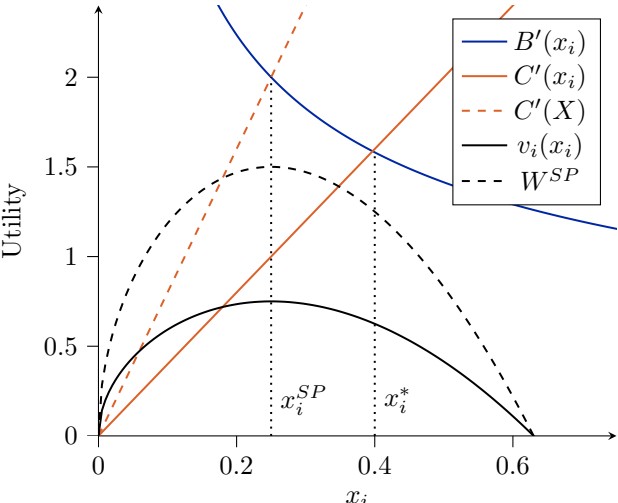

**Figure 2.** We present a simple model of public park consumption, where $B'(x_i)$ stands for the marginal benefit, $C'(x_i)$ denotes the individual optimiser's marginal cost, and $C'(X)$ indicates the marginal cost as perceived by a social planner. Additionally, $v_i(x_i)$ represents the individual's utility, and $W^{SP}$ signifies the total utility as seen by a social planner.

where $v_i$ signifies the utility from traditional income (i.e. the usage of the common), and $w_i$ encapsulates the utility from obtaining and holding the tokens. The model presented here, while a significant abstraction of reality, attempts to capture a single aspect of a complex and intricate system. It is predicated on the concept of a token that, by its very existence, offers utility by encapsulating information about a common and its use. In light of the problem at hand, the design of this token can be optimised to achieve maximum total utility.

## 2.2 A New Equilibrium

Referring back to the original social planner problem, we find that the new equilibrium can be realised with a token, provided it accounts for the external effect. In particular, it needs to consider that the cost was borne by all individuals, without them necessarily considering the increase induced by their consumption of the common resource (i.e., the externality which escalated the cost for everyone else).

To achieve the social planner's equilibrium, the token must be designed to increase the marginal cost of additional consumption $x_i$. More specifically, if it increases the marginal cost such that:

$$\underbrace{N * \frac{\partial C}{\partial X}}_{\text{Social Planner MC}} = \underbrace{\frac{\partial C}{\partial X} + \frac{\partial w_i}{\partial x_i}}_{\text{Individual MC (with token)}} \quad (12)$$

The left side of the equation takes into account the number of individuals whose consumption choice directly increases the total cost for themselves and, via the externality, also for all other participants.

Assuming the token is accepted by the community (i.e., an agreement on the common), it can provide a distinct utility. If we can

comprehend its valuation – which is likely a complex task in practice – it is feasible to instantiate a token that behaves similarly to a tax on the consumption of the good, thereby reducing the incentives for excessive consumption of the common. By introducing a secondary dimension that encapsulates information on the common, namely, a token, we construct and incorporate an additional incentive mechanism. This mechanism, in turn, can influence consumption behavior and contribute to the effective management of the common.

## 3 A Practical Application

To better illustrate and make our model tractable, consider the case of a public park in a city. The park represents a non-exclusive consumption good: people are free to visit, occupy a certain amount of space for various activities like playing, listening to music, or picnicking. They choose their desired level of consumption and derive utility from it. However, the more people visit the park, the less utility it provides. For instance, if too many people want to play football, it becomes difficult to enjoy a proper game. The more consumption that occurs in total, the less enjoyable the experience becomes. This decrease in utility is incurred by the presence of others.

### 3.1 Standard Model

Our model represents the park's total consumption as $X$ and the utility derived by an individual from consuming it as $B(x_i)$. The reduction in utility due to everyone else occupying the same common space is captured by $C(X)$. In this context, the cost of using the park (i.e., the reduction of utility it provides) is global and incurred by every individual.

For a numerical example, consider two homogeneous individuals, $i$ and $j$. The benefit of using the park is given by $B(x_i) = 0.5x_i^{-0.5}$ and the cost of using the park is given by $C(x_i) = (x_i + x_j)^2$. Fig. 2 shows the resulting marginal benefits and marginal costs. It's important to note that $x^*$ denotes the Nash equilibrium consumption of the good, where the marginal utility of consumption (solid blue line) equals the individual's marginal cost of consumption (solid red line). We see that the agents overconsume the good (i.e., the public park) as they do not account for the externalities of their consumption. Consequently, their utility is not maximized.[3]

A benevolent social planner understands the externalities and can account for them in the allocation of the good (e.g., limit park consumption by forbidding certain activities). Since their marginal cost function, represented as the orange dashed line in fig. 2, is steeper, the planner arrives at a lower individual consumption level, expressed as $x^{SP}$. The planner successfully maximizes total utility and, in turn, also maximizes individual utilities, as can be seen by the dashed black line and the solid black line respectively in fig. 2.

### 3.2 Tokenized Solution

Let us introduce a token that inversely measures the consumption of the park. With the aid of small personal computers and the Internet of Things, data about park consumption is collected. A smart contract on a decentralised ledger distributes tokens to everyone according to their consumption. If we assume that users value the

---

[3]It's worth noting that the Nash equilibrium is not Pareto optimal. A reduction in consumption would increase utility for both parties. However, this would incentivize the other player to increase their consumption even further as the total marginal cost is lower than their individual marginal cost.

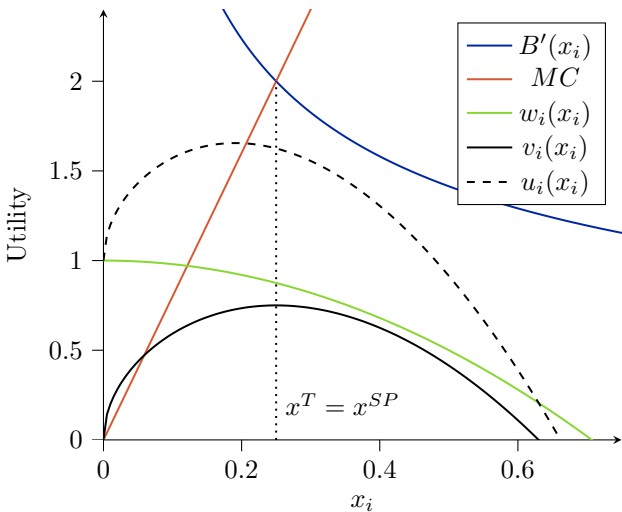

**Figure 3.** In our simple model of public park consumption, a token represents an individual's consumption. Here, $B'(x_i)$ signifies the marginal benefit, $MC$ denotes the individual optimizer's marginal cost (caused by the loss of tokens and the increment of $C$), and $w_i(x_i)$ reflects the utility derived from possessing the token. Further, $v_i(x_i)$ signifies the individual's utility derived from consumption (i.e., using the park), and $u_i(x_i)$ represents the total utility of an individual, considering the utility obtained from the token and consuming the common.

token itself—which represents the preservation of the common good (i.e., preventing the park from being overused)—it becomes possible to devise a scheme where the individual's marginal cost of consumption equals the marginal cost calculated by the social planner. As illustrated in fig. 3, if each individual initially receives a undetermined amount of tokens (valued at one unit of utility)[4], and the number of tokens they can retain–based on their consumption– is given by $w_i(x_i) = (1 - 2x^2)$, the marginal cost perceived by the individual equals that of the social planner. Consequently, a global optimum can be achieved. Fig. 3 demonstrates this, with the green line indicating the token's contribution to total utility. The decreasing function of token utility forces the individual's marginal cost (MC, red line) to mirror the social planner's marginal cost, as the loss of tokens introduces a new cost to consumption. We arrive at the optimal consumption level denoted by $x^T = x^{SP}$, which is now the same as the social planners outcome without the token. The dashed black line now represents the individual's maximisation problem. Interestingly, over consumption still exists as the compound utility $u_i(x_i)$ is not maximised; therefore, the social planner's solution will only be achieved if the social planner does not perceive the utility of the secondary token (e.g., they do not have multidimensional incentives).

As can be seen in our example, we think that if consumers value the preservation of a good, tokens which represent the consumption of a common, can be interpreted as goods that offer this product. If we assume that utility can be derived from a token and we assume that transforming information on the usage of the good can be

---

[4]The specific value of utility derived from the token is not critical to the solution. What matters is the first derivative of $w_i$.

reflected in the token, we find that it can lead to the first best solution of a social planner.

In conclusion, this example serves to illustrate the broad applicability of our model. We recognize the existence of specific problems where such a model could be particularly beneficial. For instance, common infrastructures like InterPlanetary File System (IPFS) or decentralized databases present promising areas for future research. The readily available nature of information in these systems makes them suitable candidates for exploring tokenization and incentivization strategies aimed at sustainable utilization. Additionally, it is noteworthy to mention the potential of tokenized solutions in transforming abstract ideas or concepts into marketable products. Previously, this process was either not feasible or associated with prohibitively high costs. The ability to efficiently tokenize these concepts opens up new avenues for economic innovation and value creation.

## 4 Conclusion

In this paper, we examined how the challenges of our time can be tackled through the lens of the *tragedy of the commons* and the *tokenization* thereof. We proposed a simple model of this problem, illustrating how *over consumption leads to a decrease in total utility*. We suggested a *token mechanism* that measures the usage of a common good and *incentivizes individuals* to adjust their consumption behaviour accordingly. Our model demonstrated that if consumers value the preservation of a good, tokens representing the consumption of a common good can be perceived as commodities that *promote this preservation*. Assuming that utility can be derived from a token, and that information on the usage of the good can be reflected in the token, we concluded that this approach can lead to the *optimal solution from a social planner's perspective*.

*Blockchain and distributed ledger technologies* provide a natural environment for implementing these tokens. We highlighted several reasons why tokens might be preferable to solutions involving a central planner. First, *blockchain solutions* are primarily data storage systems, which makes them ideal for storing information on the consumption of goods. Second, by design, most blockchain solutions are decentralized, aligning well with the concept of a common good. Additionally, creating tokens is cost-effective (especially when compared to non-digital assets), and *smart contracts* can manage the transformation of information.

Moreover, blockchain systems can seamlessly integrate with the *Internet of Things* and other measurement technologies needed to estimate the consumption of a common good.

In conclusion, tokens based on blockchain systems possess intriguing properties that warrant the attention of further research. For example, tokens can be designed to be non-exchangeable and thus non-marketable, a feature not common among most non-digital and digital goods. With this in mind, such tokens could offer limited exchangeability for services and goods, as exemplified in our example model (section 3.2), where payment could be made using the token befor entering the park.

As a final note and a direction for further research, we would like to highlight that *blockchain systems themselves possess properties related to the problem of the commons*. Specifically, consensus algorithms are essential because the commons, represented by the space on the blockchain, is finite, and unrestricted access would decrease its usability. Thus, mechanisms such as *Proof-of-Work*,

*Proof-of-Stake*, and *Proof-of-Authority* can be viewed as attempts to resolve this commons problem. Interestingly, the IOTA protocol uses a secondary token, *Mana*, to control access to the commons. This situation bears a striking resemblance to the problem described in our paper.

## Acknowledgments

Benjamin Kraner and Nicolò Vallarano wish to acknowledge the financial support received from the IOTA Foundation. Additionally, we extend our sincere gratitude to Olivia Saa and Luigi Vigneri of the IOTA Foundation for their invaluable contributions. The engaging discussions, insightful inputs, and constructive feedback they provided have been instrumental in shaping the direction and substance of our research. We also express our profound appreciation to Prof. Dr. Dirk Helbing and Cesare Carissimo. Their initial dialogues were fundamental in initiating and developing our ideas.

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
