# OpenReview forum: "Tokenization of the Common: An Economic Model of Multidimensional Incentives"
_ACM.org/Middleware/Workshop/DICG — DICG 2023_

### Official Review · Reviewer_sRJu · 2023-10-20
**Tokenization insufficiently justified and economic analysis out-of-scope for workshop**

**Rating:** 3
**Confidence:** 3

**Review:**

# Summary

The authors propose an economic model that uses tokens to shift the utility derived from the use of shared commons towards a better optimum. The model attempts to correct for the discrepancy between the maximization behaviour of individual agents, who increase their consumption until they no longer obtain an increase in benefit that is larger than an increase in the cost they support, and the total utility they derive from their consumption, i.e. the difference between the benefits they get and the cost they support, that happens at a lower consumption level.

# Main criticism

The main issue I have with this work is that the results are highly dependent on how the problem is formulated. Nonetheless, suppose the authors actually have formulated the problem correctly and their model sufficiently captures the dynamics of the system. Figure 2 shows that the optimum of one agent utility actually coincides with the global utility as seen from the point of view of an ideal social planner. The problem can simply be solved by communicating the individual utility function to agents so that they realize it is better for them to optimize for utility rather than simply maximizing consumption until marginal benefits equal marginal costs. This solution does not require tokens.

Now let's suppose that for a majority of problems, and contrary to the modelling of the authors shown in Figure 2, the optimum of the individual utility function does not coincide with that of the global utility function. Still, communicating the global utility function to agents allows them to optimize for that instead and additionally, each agent can monitor whether other agents are doing the same or are instead optimizing for their own (biased) individual utility to the detriment of global utility. In effect, this is already what commons achieve in practice through deliberation and self-governance processes as described in Elinor Ostrom work (already cited in [12]).

Therefore, it is not clear to me how tokens would result in better self-governance processes. Moreover, tokenization schemes carry the additional risk of introducing another set of artificial incentives that can backfire if not designed correctly and may result in agents converging towards sub-optimal utility, possibly worst than if no tokens had been introduced. To better motivate the use of tokens, the work should show how token-based incentives are actually better than deliberation and self-governance processes described in [12] that don't use tokens. Moreover, to be safe the work should argue that agent behaviour under token incentives leads to no worse optimum than if no tokenization scheme was used, when assumptions on the problem are not met.

# Relevance

Finally, the DICG workshop "focuses on the tools, frameworks, and algorithms to support the common good in a distributed environment". The paper instead attempts to provide a motivation for the use of tokens through an economic analysis and therefore appears out-of-scope for the workshop. Instead, if the economic analysis had led to insights that help understand how to better design tools, frameworks and algorithms I think it could have qualified. In its current state, I therefore recommend rejection of the paper.

---

### Official Review · Reviewer_LhsK · 2023-10-24
**Interesting proposal addressing a longstanding problem; however, it lacks key discussions, requires a more thorough evaluation, and needs a better example.**

**Rating:** 7
**Confidence:** 4

**Review:**

The paper delves into the concept of addressing the "tragedy of the commons" through an approach of tokenization. The paper uses the context of public park usage to demonstrate its practicality. The authors propose a model wherein individual consumption of a public good (the park) is regulated by a system of tokens distributed via a decentralized ledger, incentivizing individuals to self-regulate their consumption in line with the greater good. The token system mirrors the social planner's marginal cost. This design aligns individual consumption with the overall optimal level. The paper also theorizes the potential for blockchain technology to support such a system, given its properties of decentralization, data integrity, and compatibility for real-time consumption data.

Strong points:
1. **Relevance:** Addressing the tragedy of the commons is a long standing problem and highly relevant in modern society. The proposed framework is promising as it might enable other/alternative incentives as a solution.
2. **Detailed Theoretical Model:** The authors provide a detailed theoretical framework for their proposal, including clear illustrations and explanations of concepts.

Weak points:
1. **Impractical Example**: The example with the public park is a bit weird. Measuring consumption and benefit in this context is challenging. Unlike resources such as water or energy, the overconsumption of space in a public park is intangible and subjective. Implementing a token system would require constant monitoring of individual usage of the park, which raises practical questions about how this would be achieved. Would it be based on time spent, area occupied, or some other metric? Also, to monitor individuals in public spaces raises significant privacy concerns and logistical challenges.
In other examples of the tragedy of the commons, like overfishing or air pollution, the negative consequences of overexploitation are substantial and relatively immediate. In the case of a park, beyond the discomfort of overcrowding, the repercussions of overuse are less dire and immediate, potentially weakening the perceived necessity for a token system.

2. **Paper Clarity**. The paper would benefit from a restructuring. For example, Figure 1 is not explained in the text, and thus I'm missing the full inspiration and the parallels with your solution/model.

Improvement suggestions for this and/or extended version of the paper:

- Consider a different practical example. It would be nice to consider an example closer to Web3, ideally with some empirical data. It can be some common good such as  Web3 infrastructure (Blockchain, IPFS), or some common funds in DAOs.
This way you can compare your solutions against a popular approach of Quadratic Funding/Voting (https://vitalik.ca/general/2019/12/07/quadratic.html).
For instance, how does the token model, in promoting responsible consumption, compare to the preference expression in Quadratic Voting? Does the token model ensure resource sustainability more effectively than Quadratic Funding's method of resource allocation?

- Add more details on the tokenomics part. Discuss the mechanics of token distribution, supply, demand, transferability, and design features that incentivize desired behaviors.

---

### Official Review · Reviewer_z8pj · 2023-10-31
**Interesting idea, can be accepted with some revisions**

**Rating:** 7
**Confidence:** 3

**Review:**

The proposed manuscript provides some interesting insights into the instrumentalization of novel token-based incentive mechanisms to address some instantiations of a tragedy of the commons. The authors provide a formal model to represent an interaction between agents competing for shared resources, where tokens also carry an information signalling function about resources and their use. The provided analysis does carry an interesting observation that this signalling function may constitute an additional incentive mechanism that influences consumption behaviour and decreases the risk of overconsumption.

The manuscript also proposes a practical illustration of this mechanism, applying the tokenization mechanism for common park use. Admittedly, the application of the provided analysis to this specific use case is a weaker point of the manuscript. Even though it is just an illustrative use case, it does not present the explanatory affordances and limitations of the proposed model in the best light. There is a noticeable gap between the level of abstraction of the model and the semantics of the use case, both in terms of problem definition and (arguably) feasibility of the proposed solution.

Nevertheless, all in all, the manuscript does fit well in the scope of the workshop as a clear theoretical contribution indicated in the CFP. I believe it does carry the potential to spark an interesting discussion regarding the application of tokenization mechanisms and incentives in the context of the tragedy of the commons. However, the proposed use case does undermine the quality of the contribution. Thus, I would like to urge the authors to consider revising the manuscript with a different use case in order to provide better matching between the formal analysis and illustrative semantics.